# Business sustainability of medicinal plant production under risk in the northwest region of Bangladesh: A simulation analysis

**Md. Abu Saiyem**[1☉], **Mst. Fatema Begum**[2‡], **Mst. Esmat Ara Begum**[3‡], **Mohammad Ismail Hossain**[4☉]*

**1** Department of Agricultural Extension under Ministry of Agriculture, Dhaka, Bangladesh, **2** Centre for Innovation Studies, Dhaka, Bangladesh, **3** Tuber Crops Research Centre, Bangladesh Agricultural Research Institute, Gazipur, Bangladesh, **4** Department of Agribusiness and Marketing, Bangladesh Agricultural University, Mymensingh, Bangladesh

☉ These authors contributed equally to this work.
‡ MFB and MEAB also contributed equally to this work.
* ismailho12@yahoo.co.in

## Abstract

Medicinal plants (MP) provide an opportunity for profitable farming, agricultural diversification, and sustainable livelihoods in Bangladesh. This study aims to examine the business sustainability of MP production under risks, i.e., price, yield and market absorption. Data were collected from 196 individual farms that cultivate *Aloe vera (L), Bombax ceiba (L)* root, and *Withania somnifera (L)*. The business viability analysis was carried out by evaluating the net present value (NPV), internal rate of return (IRR), benefit-cost ratio (BCR), return on investment (ROI), and real option value (ROV) of selected MP. The results indicated that the NPV was positive, the BCR was greater than 1, and the IRR exceeded the prevailing bank rate. It also found that the production of MP faces three major risks: pricing, yield, and market absorption. The simulation analysis's findings indicated that MP producers will eventually experience losses with embedded risks. On the other hand, the ROV finding indicated that NPV increases with option value over time. Farmers may receive profit from their investments in *A. vera, B. ceiba* root, and *W. somnifera*, and their farming business will eventually become sustainable in the long run. Thus, farmers should plan for producing MP in the long run and policy should be triggered on tackling risk reduction strategies through introducing marketing contract.

## Introduction

The majority of the population in developing and developed countries (about 70% and 65%, respectively) relies on herbal medicine for primary health care and medicine [1–6]. About 25% of modern medicines are manufactured by MP [7,8]. Global sales of herbal medicines were estimated at $151.91 billion in 2021 [9] and are

**Data availability statement:** All relevant data are within the manuscript and its Supporting information files.

**Funding:** The author(s) received no specific funding for this work.

**Competing interests:** The authors have declared that no competing interests exist.

expected to grow to $347.5 billion in 2029. During the COVID-19 pandemic, demand for medicinal herbs has increased in all countries compared to pre-pandemic levels. Asian countries account for 42% of global market share, 50% of exports, and 45% of global sales [10]. In Bangladesh, the annual market sale of medicinal plants is about 3 billion BDT, and the annual consumption is about 20 tons [11]. Additionally, a quarter of pharmaceutical drugs in Bangladesh are manufactured from MP [11–13]. Besides this, MP cultivation is an important source of income [14,15], increases food security [16], meets nutritional needs and provides various health resources [17,18]. In fact, the economic value of MP is so valuable that it exceeds statistics and prudence [16,19,20], and there is a need for trade in formal and informal markets [21].

Although wild-grown MP is generally considered more potent than cultivated MP, the scarcity of wild resources has accelerated the commercial cultivation of MP in recent years [22–24]. Training on MP production approaches provides an opportunity to introduce new skills, new expectations, and new business opportunities [25–27]. Also, MP cultivation in cereal crop cultivation land has been identified as an effective alternative to conventional crops [1,5,28]. However, the long-term profitability of the MP is unclear. Many farmers make decisions based on short-term profits, which is common in developing countries. Furthermore, farmers in developing countries do not keep records. The realization that actions taken today can have long-term consequences raises new challenges for farmers' decision-making. In the current situation, the challenge is to find a clear picture of the long-term viability of MP production. Liontakis and Tzouramani [29] and Saiyem et al. [5] described the risks associated with MP production because MP production and prices threaten farm incomes and are inherently volatile, which should be considered in decision-making options [9,30]. The market price of MP is also unstable and fluctuates significantly from season to season. Thus, to achieve higher returns from MP production, farmers need to adopt a wide range of rational risk management practices. One possible option is to explore possible options by considering different production scenarios, taking into account the risks. However, not all MP farmers pursue short-term profits but continue production with a long-term perspective in mind. Therefore, it is important to assess whether investments in MP production can increase future premiums for farmers in Bangladesh and make MP production sustainable.

There is an extensive literature on MP production, which generally focuses on profitability [1,6,31], produce risk strains [5], and differences in reproductive constraints [32,33], value chain mapping and management [28,34–40], conservation and utilization of resources [41–47], and their sustainability [29,48–51]. Given the risks of MP production, limited research has been conducted on business sustainability. Only Galleria et al. [50] conducted research on the economics of production and marketing of important medicinal and aromatic plants, where the authors applied financial tools (NPV, BCR, and IRR) to assess the economic profitability of crop farming. In 2016, Liontakis and Tsouramani [29] demonstrated the economic sustainability of organic *A. Vera* cultivation under the conditions of embedded risks and uncertainties and estimates the level of sustainability by incorporating risk into the profit equation. Saiyem et al. [14]) also studied the profitability options of MP under risk by applying profit

equations and found that farmers experienced negative return when considering risks (price, yield, and market absorption). However, NPV, BCR, IRR and simulated returns are indicators of sustainability, but their results can have long-term implications [52–54]. With keep this in mind and taking into account the existing literature, this study is conducted to discover the business sustainability of risky MP production. The aim of this study is to estimate the level of investment in MP production which increase the profitable value of future premiums and enable Bangladeshi producers to sustain production.

We believe that the results of this study will benefit farmers and researchers and help decision makers to make appropriate production decisions, future research and policy development.

## Materials and methods

### Conceptual framework

This study was conceptualized for business sustainability of MP production. Farm-level business sustainability means that MP farms can be managed in a way that ensures long-term returns and limits risk. One of the most important decisions made by a business farm is capital budgeting. Capital budgeting involves long-term investment commitments, a degree of uncertainty, and plays an important role in the investment decision-making process [55]. To assist the investment evaluation process for MP production, the discounted cash flow techniques such as NPV analysis can be used. Also, MP production, like any other crop, involves risk, and farmers often face difficulty in estimating the expected yields. Thus, simulation modeling was used to incorporate risk and uncertainty into probabilistic estimates of $\widetilde{NPV}$. In this case, the variables describing risk and uncertainty are entered into the probabilistic rather than deterministic model. The MP farmers face three significant risks: price uncertainty, market absorption, and lower yield risks, described in Fig 1. The risks are included in the simulation model of $\widetilde{NPV}$ to observe the future return trends. According to Dixit and Pindyck [54], the NPV method assumes an irreversible investment, meaning that if a farm decides to invest, it can forgo the option of waiting for new information. Therefore, in this study, to avoid doubts about the NPV method, a real options approach has been applied to model a decision-making process that is considered dynamic over time. It ultimately showed how the expansion of MP production depends on the long-term profitability and sustainability of the business (Fig 1).

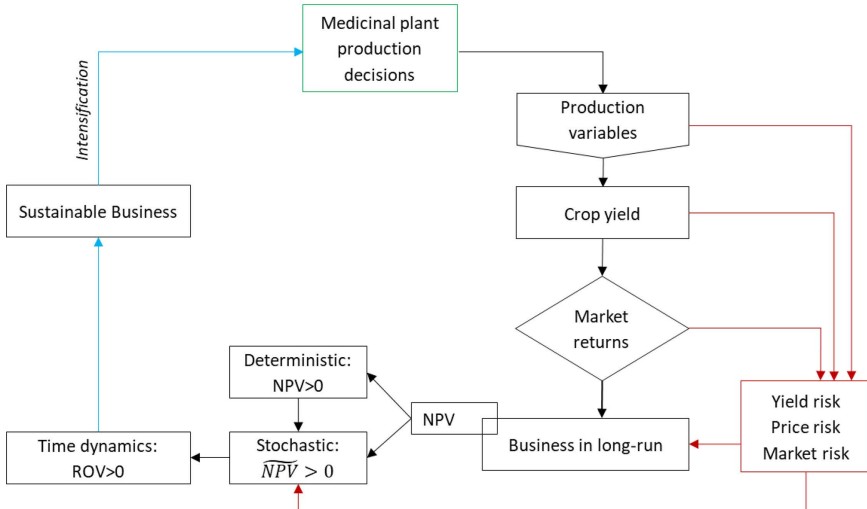

**Fig 1. Conceptual framework of the study.**

## Selection of the medicinal plants

More than 38,660 MP species are native to Asia, and about 78 species are commercially cultivated [56–58]. In Bangladesh, there are six species called *A. vera, B. ceiba, W. somnifera, Asparagus racemosus, Scoparia dulcis,* and *Rauwolfia serpentina* that are commonly cultivated by farmers [28,59]. Among the commercial species, *A. Vera, B. ceiba* and *W. somniferous* (locally known as *Aloevera, Shimulmul* and *Ashwagandha*) are widely cultivated in Bangladesh (especially in Natore District, popularly known as medicinal region), covering an area of 266 hectares [60]. Each of these plants is perennial and its leaves and roots contain many biological attributes [61–65]. Among the representatives, about 38% of the area is covered by *A. vera*, where *B. ceiba* species covered 27%, and *W. somnifera* species covered 16% [66]. Therefore, these three MP were selected for this study. The perennial plant *A. vera* generally does not reproduce by seed. It is regenerated from the pulp. *A. vera* plants can be harvested by removing 3–4 leaves from each plant every 6–8 weeks. On the other hand, *W. somnifera* is a hardy, drought-tolerant plant whose roots, seeds, and leaves are used for medicinal purposes. The roots are dug and harvested 150–180 days after sowing. Another MP, *B. ceiba* is often known as the "silent doctor" due to its medicinal properties. *B. ceiba* plants can be harvested after 8–10 months when mature plants reach a height of 0.61–1.22 m.

## Selection of the research area

Commercial cultivation of MP species has been recorded in *Rajshahi, Chattogram* and some parts of hilly regions of Bangladesh. The Natore and Bogura regions are popular with commercial practitioners who cultivate the MP at their crop land. *A. vera* is cultivated as one of the most important crops in these regions [5,28,67,35,68]. However, *B. ceiba* and *W. somnifera* are widely cultivated in Bogura and Natore regions [5,9,28]. Bogura Sadar and Natore Sadar upazila have unique identities for MP production. Therefore, these two upazilas were selected for this study. Regarding climatic conditions, the selected two regions' climate is also suitable for MP cultivation but vulnerable to drought, and the mean annual maximum temperature has increased by 0.16°C over the past 20 years [5].

## Selection of sample, sample size and sampling techniques

A multi-stage purposive sampling method was used to select the study areas. In the first stage, two widely MP cultivating areas, are *the* Natore and Bogura districts (1st sub-administrative units) under *the* Rajshahi division (administrative units), were selected. *Within* these two districts, the Natore Sadar and Bogura Sadar upazilas (2nd sub-administrative units) were selected, which consist of 11 unions (3rd sub-administrative units) and 4 unions, respectively, known for the area of commercial cultivation of MP [5,28,67]. Then, the list of MP farmers of those unions in the two upazilas was collected from the respected Upazila Agricultural Officer (UAO). Finally, MP farmers were randomly selected from these lists. To estimate the appropriate sample size, this study used the formula (Equ1) adopted from Kothari [69].

$$n = \frac{Z^2 * \sigma^2 * N}{(N-1)e^2 + Z^2 * \sigma^2}$$

(1)

Where, n = sample size, N = population size, σ = estimated population variance; e = desired precision, Z = standard normal deviation, e = marginal error. According to BBS regional statistics [70], Natore Sadar and Bogura Sadar upazila have approximately 56097 and 33587 farms, respectively. Among those about 5,000 farms in these two upazilas produce MP, of which 30% farms (about 1,500 farms) commercially grow different MP [71,72]. From the 1,500 commercial MP growing farms, about 65% (approximately 960 farms) are cultivate the selected three MP (*A. vera, B. ceiba* and *W. Somnifera*). Therefore, the population size (N) was considered at 960 farms. Due to time and economic constraints, the margin of error was set at 10%, the confidence level was fixed at 95%, and the population variance was assumed to be 75%. Based on the values (N = 960, e = 10%, Z = 1.96, σ = 75% of the population variance) the estimated sample size was calculated to be 176. However, a slightly larger sample of 196 was considered for this study. Data were collected from these 196

households cultivating the selected three MP (*A. vera*, *B. ceiba* and *W. somnifera*) using the face-to-face interview method through a pre-tested structured interview schedule.

Proportion of *A. vera*, *B. ceiba* and *W. somnifera* farmers in the study area recorded 63%, 16%, and 21%, respectively [60]. Therefore, the sample was distributed by using the Probability Proportionate to Size (PPS) method [73]. Finally, the sample size of *A. vera* was 123 (77 from Natore, 46 from Bogura), 31 for *B. ceiba* (19 from Natore, 12 from Bogura) and 42 for *W. somnifera* (26 from Natore and 16 from Bogura).

Primary data were collected from farmers through survey method from December 2018 to July 2020 by using a structured questionnaire. The questionnaire is divided into five sections: socio-economic, production, marketing, embedded risks and constraints. The socioeconomic section of the questionnaire is consisting of elements related to (1) information about the respondent, (2) land holding and tenure (3) household assets and income, and (4) information related to training and financing. The second section is cultivation gathered information regarding (1) cultivation of medicinal plant, (2) cropping pattern and (3) material used for MP cultivation including labor requirement and costs. Marketing is linking with cultivation section contain all market related information together with contract marketing. The fourth section gathered the list of problems or constraints for cultivation, harvest, post-harvest and marketing activities of medicinal plant. The last section is about risks embedded with production and marketing practices.

Most of the data obtained from questionnaire survey consist of continuous data, but only the data related to constraints in fourth section is developed according to a five-point likert scale. Some relational variables are built in order to know the relational marketing of the medicinal plant susb-ector.

STATA (version 14.0), R (version 4.2), and @riskAmp (free version) statistical software and the Solver Plugin (for Microsoft Excel 2019 version) were used for data analysis.

## Ethical approval

Because all human involvement was guaranteed in accordance with local laws and institutional concerns, this study exempt from ethical approval. Besides, this study was carried out in accordance with the Helsinki Declaration's guidelines for human subjects research. Additionally, since there was no institutional ethical board to authorize the social science study during the data collection periods, there was no opportunity to apply for Research Ethics Board (REB) approval. Due to this academic limitation and local legislation, ethics approval was not required for the study.

The study relied on publicly available data, secondary sources, and information voluntarily provided by farmers through informal interviews and field visits. As such, the scope of the research did not require prior ethical clearance under existing national research guidelines for non-clinical, economic studies. Nevertheless, all activities were conducted with respect for local norms, with verbal consent obtained from participants, and in alignment with the ethical standards for socio-economic research. We are committed to upholding high ethical standards in all aspects of our research.

## Consent to participate

Verbal consent was obtained from each respondent during the data collection period (in-person interview), as the majority of them lack literacy and are reluctant to sign due to the possibility of tax or liability implications. However, we cleared them all, and verbal consent was given by the responders. We documented this in the paper-based questionnaire by putting a tick during face-to-face data collection time (the top page of questionnaire is attached as supporting information 1 for further details) and this wasn't approved by Institutional Research Board due to academic limitations. In this ground, this study received a waiver of written consent from the respondents. No minors were involved in this study.

## Analytical methods and algorithms

The analysis algorithm in this study consists of three parts: (i) NPV estimation, (ii) incorporating risk into the NPV, and (iii) determining ROV dynamics over time. At first, NPV is estimated as a discounted cash flow method to support the

investment evaluation process, where NPV can be defined as the sum of the annual cash flows discounted for the year in collecting capital costs, minus the investment outlay [74,75]. Denoting the expenses in year t as $c_t$ and the revenues as $b_t$ then NPV is formulated as follows (Equ 2):

$$NPV = \sum_{t=1}^{T} \frac{b_t}{(1+r)^t} - \sum_{t=1}^{T} \frac{c_t}{(1+r)^t} - I_0$$

(2)

Here, $I_0$ is the initial investment, T is the 10-year planning period in the current analysis, and r is the discount rate. The Equ 2 produced the NPV results, which are highly dependent on the discount rate chosen. Thus, an appropriate discount rate must be chosen carefully because very high discount rates can result in a zero or negative NPV [76]. The process of converting future amounts into present is called "discounting," and its element is the discount factor (DF). This is an important part of the time value of money, which is calculated by adding the discount rate to one which is then raised to the negative power of several periods. Mathematically, discount factor (DF) is expressed as follows (Equ 3):

$$DF = (1 + \frac{i}{n})^{-n*t}$$

(3)

Where i = discount rate, t = number of years, n = number of compounding periods of the annual discount rate. In the discount concept, $\frac{b_t}{(1+r)^t} - \frac{c_t}{(1+r)^t}$ is called free cash flow. Therefore, the above Equ 3 can be written as (Equ 4):

$$NPV = -I_0 + \sum_{t=1}^{T} \frac{FCF_t}{(1+r)^t} + \frac{E}{(1+r)^T}$$

(4)

Here, $FCF_t$ = free cash flow in year t (t = 0, 1, 2, … n); E = final value; r = discount rate. Typically, investments with a positive NPV are considered profitable, while investments with a negative NPV result in a net loss. The assumptions for measuring NPV are summarized in Table 1

Besides NPV, the study also examines the cost-benefit ratio (BCR), internal rate of return (IRR) and return on invested capital (RIC), following Conrad [77], who described BCR is simply the ratio of the present value of benefits ($\sum_{t=1}^{T} \frac{b_t}{(1+r)^t}$) to the present value of costs ($\sum_{t=1}^{T} \frac{c_t}{(1+r)^t}$), where, in IRR, $\sum_{t=0}^{T} p^t B_t = \sum_{t=0}^{T} p^t B_t$, and in RIC, no balance is available for reinvestment. These apply to the NPV criterion, which provides exactly the same information and is transformed in the same way.

In the second layer, risks are included in the NPV equation (Equ 4). Selected MP (*A. vera*, *B. ceiba* root, and *W. somnifera*) production is characterized by several underlying factors that strongly influence profitability when production takes place under uncertainty. Fluctuations in yield were found to be a key source of risk, since variations in soil fertility, water availability, and pest pressures often caused inconsistent harvests—a challenge also observed for *A. vera* and other MPs across South

**Table 1. Key assumptions used in Net-Present-Value analysis.**

| Indicators | Unit | Value |
|---|---|---|
| Farm size | Hectare | 1 |
| Ownership | % | 100 |
| Loan length | year | 1 |
| Flat rate of interest | % | 9 |
| initial equity | BDT | 0 |
| Life cycle | year | 10 |
| Discount rate | % | 8 |

Asia [29,78]. Price volatility further undermined producer confidence, as marketing systems remain largely unstructured and farmers are heavily dependent on intermediaries, a situation mirrored in studies from Bangladesh and Nepal [38,79]. The high and unstable cost of quality planting materials added to production uncertainty, particularly in the case of *B. ceiba* and *W. somnifera*, where limited access to seedlings has constrained farm-level expansion [4,14]. Moreover, weak market absorption capacity often left producers vulnerable to distress sales during periods of oversupply, reflecting the absence of adequate processing facilities or contractual arrangements [21,35]. Taken together, these intertwined risks emerged as the defining constraints on the long-term business sustainability of medicinal plant enterprises in Bangladesh. Thetrefore, the study constructed four stochastic variables (Table 2), which are distributed as GRKS (Grey, Richardson, Klose, and Schuman) distribution, and simulated subjective probability distributions based on minimal input data [80]. The minimum (min), median (mid), and maximum (max) values are used here. GRKS is useful when minimal information about the distribution is available, requiring only the minimum, mean, and maximum values as limits [80]. The GRKS distribution assumes that 50% of the observations will be greater than the mode value, 2.28% of the values will be greater than the maximum, and 2.28% will be less than the minimum, with the minimum probabilities to occur are 10% each.

Stochastic risk models have been approached by the study by assigning probability distributions to specific exogenous and endogenous variables, though it is technically complex to construct and difficult to test or falsify [81]. Considering the recommendation of Asci et al. [82] on risk-taking decision-making in agricultural management, the Monte-Carlo simulation method was used as a stochastic model in this study. This method attempts to accurately address the complexity of real-world situations [83]. Monte-Carlo simulations consider all possible outcomes, where the output is not a single value but a continuous range of possible values with associated probabilities representing the NPV (<0 or >0) range for agriculture. This simulation equation of NPV is expressed as (Equ 5):

$$\widetilde{NPV}_0 = -I_0 + \sum_{t=1}^{T} \frac{\widetilde{FCF}_t}{\left(1 + r\right)^t} + \frac{E}{\left(1 + r\right)^T}$$

(5)

Here, $\widetilde{NPV}_0$ represents the simulated net present value and $\widetilde{FCF}_t$ represents the simulated free cash-flow. The cash-flow is stochastic with respect to income, and price and income absorption rate are dynamic and used as risk variables. Thus, the profit-flow ($\widetilde{BF}_i$) is estimated using the Monte Carlo simulation method as follows (Equ 6):

$$\widetilde{BF}_i = \sum \left( \widetilde{Y}_{ij} * \widetilde{p}_{ij} * \widetilde{m}_{ij} \right)$$

(6)

And, the cost flow ($\widetilde{CF}_i$) is estimated as follows (Equ 7):

$$\widetilde{CF}_i = \left[ (X_{si} * \widetilde{p}_{si}) + \sum_{j=2}^{n} VC_{ij} \right] - \sum_{j=1}^{n} FC_{ij}$$

(7)

**Table 2. Distribution of risk variables used in the simulation model.**

| Risk variable | Notation | Unit | Type of distribution |
|---|---|---|---|
| Yield | $\widetilde{Y}_{ij}$ | kg/ha | GRKS distribution: (Min, Max, Mid♯) |
| Producer's price | $\widetilde{p}_{ij}$ | BDT/kg | GRKS distribution: (Min, Max, Mid) |
| Seed price | $\widetilde{p}_{si}$ | BDT/unit | GRKS distribution: (Min, Max, Mid) |
| Market Absorption | $\widetilde{m}_{ij}$ | % | GRKS distribution: (Min, Max, Mid) |

♯GRKS parameters are Min (minimum), Mid (mid-point or middle or mode), and Max (maximum).

Where, $\widetilde{Y}_i$ = Stochastic yield, $\widetilde{m}_i$ = Stochastic percentage of yields absorbed by marketing channel, $\widetilde{p}_i$ = stochastic price in the marketing channel, $X_{si}$ = quantity of seedlings, $\widetilde{p}_{si}$ = stochastic price of seedlings, $VC_{iJ}$ = cost of other variable items, and $FC_{iJ}$ = cost of fixed items. By introducing stochastic input parameters into the simulation model, the results of $\widetilde{NPV}$ are stochastically estimated as equ-5 with 1000 iterations. The study was also analyzed to characterize the outcome $Y (= \widetilde{NPV})$, considering N samples by estimating the mean value of $Y (= \widetilde{NPV})$, their variance, reliability, probability of failure, probability density function (*pdf*), and cumulative density function (*cdf*). Given the values, use the corresponding *pdf* plot and *cdf* plot to summarize and compare the simulated probability distribution of NPV results. While computing the simulation statistic values, the study statistically tests the normality of the distribution using the Jarque-Bera (JB) test [84] and the Shapiro-Wilk (W) test [85] statistics.

The third step uses a multiplicative stochastic procedure to calculate the real option value (ROV). ROA considers the future value of agricultural investments in current investment decisions, determines investment irreversibility, models dynamic decision-making processes, and allows for flexibility in agricultural investments, including non-linear distributions of cash flows or recent changes in risk profiles. Following Asci et al. [82], the study calculated the ROV by using the binomial decision tree procedure described by Copeland and Antikarov [86] and used by Iwai and Emerson [87], where the analytical model is called the binomial option pricing model. This model is currently the most widely used method for valuing real options. The binomial option pricing model describes price movements over time in which the value of an asset can move to one of two possible prices, depending on the associated probabilities. Fig 2 shows the process of a binomial option pricing model through a decision tree, which gives a clear idea of how a decision tree works.

In the binomial option pricing model, a tree consists of a series of nodes and branches [82], where each node has upper value and a lower value of NPV, which are calculated by following formula (Equs 8 and 9):

The upper value (uNPV) for node t is:

$$NPV_t = PV_{t-1} * e^{(\sigma_z \sqrt{dt})} \tag{8}$$

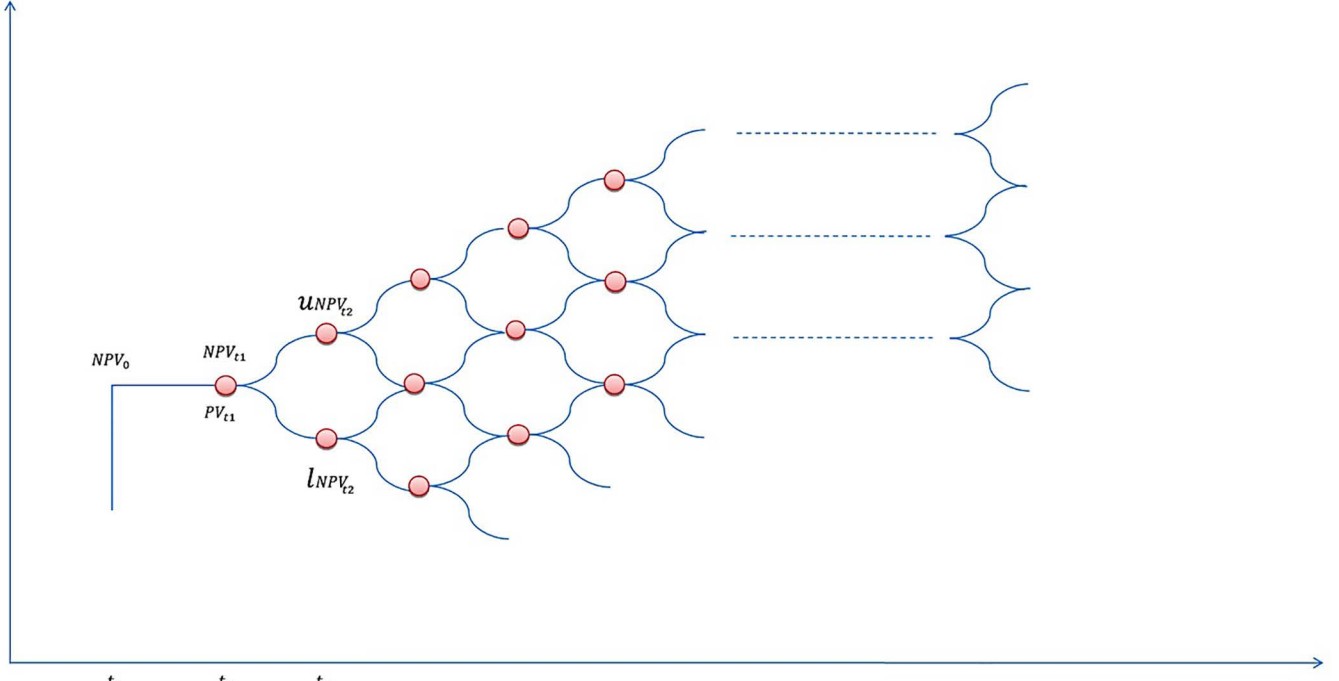

**Fig 2. Binomial decision tree.**

The lower value ($l$NPV) for node t is:

$$NPV_t = PV_{t-1} * e^{\left(-\sigma_z \sqrt{dt}\right)} \tag{9}$$

Where $\sqrt{dt}$ = time difference and $\sigma_z$ is volatility. Volatility $\sigma_z$ is defined as the standard deviation of of z, i.e., (Equ 11).

$$\sigma_z = std.dev\,(z) \tag{10}$$

Here, z is the logarithmic ratio. During each simulation test, the value of future cash flows is estimated for two periods: the first period of time and the present time. The cash flows are discounted and summed at points of time 0 and 1, and then the logarithmic ratio (z) is calculated as follows (Equ 11):

$$z = \ln\left(\frac{PV_1 + FCF_1}{PV_0}\right) \tag{11}$$

Where, $PV_1$ and $FCF_1$ means the present value and free cash flow at time t (= 1), and $PV_0$ means project's present value at the beginning of the project at time t (= 0). At any time of x, the present value can be calculated as $PV_x = \sum_{t=x+1}^{n}\left(\frac{FCF_1}{(1+WACC)^{t-1}}\right)$, where $WACC$ is the weighted average cost of capital.

Therefore, ROV is calculated as a reverse induction procedure from the expiry to today according to the following formula (Equ 12).

$$ROV\ :\ E_0 MAX\,(at\ t = T)\,[0,\ (V_t - X)] \tag{12}$$

Here, $V_t - X$ represents the comparison of possible values to select the best of the possible alternatives. ROV analysis uses the expectation of maximum value when making a decision after information is revealed (maximized at $t = T$). Finally, the study calculates NPV with option value by simply adding the option value to the static NPV value. The study also statistically tests the normality of the distribution before calculating the simulation statistic using the Jarque-Bera (JB) test [84] and Shapiro-Wilk (W) test [85] statistics.

## Results and discussions

It is observed during the survey that *A. vera* farms range from 0.01 to 0.41 hectares. The detected *A. vera* cultivar is a local variety. It is a perennial plant and normally not propagated through seeds. The average planting density ranged from 52393 to 99787 plants per hectare. The planting months ranged from November to April. It is also detected that in all surveyed *A. vera* farms, the manual farming system for planting, weeding, and harvesting is utilized. The furrow-irrigation system is used also. The average economic life of an *A. vera* plant is 10–15 years.

In the case of *B. ceiba*, the surveyed farms cover an area ranging from 0.004 to 0.5 hectares. Local varieties are the cultivars detected in all cases, and seed is used in propagation. The average quantity of seed used in the field ranged from 74.8 to 131.6 kg per hectare. The production cycle of *B. ceiba* is roughly 11 months, and the seedling months range from November to April. For all farms the farming system is found to be manual, and a flood-irrigation system is utilized.

The *W. somnifera* farms identified in the study area covered an area of 0.02 to 0.27 hectares. The seedling period of plants is between November and April. The average quantity of seeds is 10.3 to 56.1 kg per hectare, with local varieties being the predominant cultivar identified. Flood irrigation is utilized. Plants have a tree with a maximum height of 1.5 meters at the maturity stage. It should be noted that, as farmers reduced supply channels by direct sale of produce to local traders (rather than to wholesalers), farmers have a greater contracting power in the course of the price negotiations; the average market

price (farm-gate) for these farms is slightly higher. Regarding product commercialization, most of the farmers sold dry seeds, roots, and shots directly to local medicinal plant traders.

## The Net-Present-Value of medicinal plant production

Table 3 showed the NPV results of *A. vera* production. The NPV at the end of 10$^{th}$ year was found to be positive at a discount rate of 8%. This means that the investment made in the farming business of *A. vera* production is financially feasible. The farmers would earn BDT 1009546.16 after 10$^{th}$ year from the present investment. BCR is observed to 1.17, which is more than one, indicating that farmers can earn BDT 1.17 by investing BDT 1.00 at the end of 10$^{th}$ year. IRR is 22.58% which is more than the discount rate of 8%, that clearly indicates that investment in the farming business of *A. vera* production is profitable. A MP production farm must satisfy the rule $r(= IRR) > \delta$ to justify the business that start today. The internal rate of return will not be unique, and unable to solve the questions about the availability of positive balances, prior to $t = T$, and whether these balances have re-investment options. Return on investment capital (RIC) can answer this question. To find the RIC, it is need to define a project balance in $t = \tau$ as $PB_t = \sum_{t=0}^{T} N_t (1 + i_t)^{\tau-t}$ where the rate $i_t$ will be either the risk-free discount rate $\delta$, or the RIC according to the rule of (i) $PB_t = (1 + RIC)PB_{\tau-1} + N_t$ if $PB_{\tau-1} < 0$, or (ii) $PB_t = (1 + \delta)PB_{\tau-1} + N_t$ if $PB_{\tau-1} > 0$. Conrad [72] defined the project balance in recursive form and the presumption is that if the project's balance in $\tau - 1$ is negative, no balance is available for re-investment, and the RIC is the appropriate marginal return. If the project's balance in $\tau - 1$ is positive, that balance is invested for one period at the risk-free discount rate $\delta$. Thus depending on the project's balance in $\tau - 1$, that balance will be compounded forward to $\tau$ by either the RIC (if $PB_{\tau-1} < 0$) or the risk-free rate $\delta$ (if $PB_{\tau-1} > 0$). At this point, however, the RIC is unknown, Solver has been used to drive the value of $PB_t$ to 0 to changing the value from initial guess of 0.08. Solver quickly finds the real RIC and found to be 0.22581, is close to the IRR = 0.22580 and more than $\delta = 8\%$, which also clearly indicated the profitable option of investment on the farming business of *A. Vera*. From the Table 3 it could be noted that at t = 10, the IRR and RIC are not significantly different. Table 4 showed the results of *B. ceiba*. The NPV was BDT 1001362.37, BCR was 1.35, and IRR was 43.14%. These indicated that investment in *B. ceiba* production was profitable in long-run. With regard to the opportunities

**Table 3. Cash-flow streams and estimation of NPV, BCR, IRR and RIC for *A. vera* production (BDT//ha).**

| t | Bt | Ct | Nt | Discount Factor | Discounted Bt | Discounted Ct | Discounted Nt | Nt at IRR | PBt-1 | (1 + ?)*PBt-1 | PBt |
|---|---|---|---|---|---|---|---|---|---|---|---|
| 1 | 0 | 1116867.2 | −1116867.15 | 0.9259 | 0.00 | 1034136.25 | −1034136.25 | −911126.34 | 0.00 | 0.00 | −1116867.15 |
| 2 | 500183.43 | 782857.57 | −282674.14 | 0.8573 | 428826.67 | 671174.19 | −242347.51 | −188122.29 | −1116867.15 | −1369066.15 | −1651740.29 |
| 3 | 1327523.03 | 863525.67 | 463997.36 | 0.7938 | 1053830.58 | 685494.52 | 368336.06 | 251910.79 | −1651740.29 | −2024718.62 | −1560721.26 |
| 4 | 1327523.03 | 863525.67 | 463997.36 | 0.7350 | 975769.06 | 634717.15 | 341051.91 | 205505.69 | −1560721.26 | −1913146.65 | −1449149.29 |
| 5 | 1327523.03 | 863525.67 | 463997.36 | 0.6806 | 903489.87 | 587701.06 | 315788.81 | 167648.99 | −1449149.29 | −1776380.69 | −1312383.33 |
| 6 | 1327523.03 | 863525.67 | 463997.36 | 0.6302 | 836564.69 | 544167.65 | 292397.04 | 136765.96 | −1312383.33 | −1608731.70 | −1144734.34 |
| 7 | 1327523.03 | 863525.67 | 463997.36 | 0.5835 | 774596.94 | 503858.93 | 270738.00 | 111571.97 | −1144734.34 | −1403226.01 | −939228.65 |
| 8 | 1327523.03 | 863525.67 | 463997.36 | 0.5403 | 717219.39 | 466536.05 | 250683.34 | 91019.03 | −939228.65 | −1151315.23 | −687317.87 |
| 9 | 1327523.03 | 863525.67 | 463997.36 | 0.5002 | 664092.02 | 431977.82 | 232114.20 | 74252.19 | −687317.87 | −842520.64 | −378523.28 |
| 10 | 1327523.03 | 863525.67 | 463997.36 | 0.4632 | 614900.02 | 399979.47 | 214920.56 | 60574.01 | −378523.28 | −463997.36 | 0.00 |
| Σ | | | | | 6969289.24 | 5959743.09 | 1009546.16 | 0.00 | | | |
| NPV | 1009546.16 | | | | | | | | | | |
| BCR | 1.169 | | | | | | | | | | |
| IRR | 0.2258 | | | | | | | | | | |
| RIC | 0.22581 | | | | | | | | | | |

*Source: Calculation was based on equ-2, 3 & 4.*

**Table 4. Cash-flow streams and estimation of NPV, BCR, IRR and RIC for *B. ceiba* production (BDT./ha).**

| t | Bt | Ct | Cash balance | Nt | Discount Factor | Discounted Bt | Discounted Ct | Discounted Nt | Nt at IRR | PBt-1 | (1+?)*PBt-1 | PBt |
|---|---|---|---|---|---|---|---|---|---|---|---|---|
| 1 | 0 | 598702.0 | −598702.04 | −598702.04 | 0.9259 | 0.00 | 554353.74 | −554353.74 | −418252.52 | 0.00 | 0.00 | −598702.04 |
| 2 | 674064.17 | 405102.5 | −329740.37 | 268961.67 | 0.8573 | 577901.38 | 347310.10 | 230591.28 | 131264.13 | −598702.04 | −857004.11 | −588042.44 |
| 3 | 674064.17 | 405102.5 | −60778.70 | 268961.67 | 0.7938 | 535093.87 | 321583.43 | 213510.45 | 91700.96 | −588042.44 | −841745.57 | −572783.90 |
| 4 | 674064.17 | 405102.5 | 208182.97 | 268961.67 | 0.7350 | 495457.29 | 297762.43 | 197694.86 | 64062.18 | −572783.90 | −819903.93 | −550942.26 |
| 5 | 674064.17 | 405102.5 | 477144.64 | 268961.67 | 0.6806 | 458756.75 | 275705.95 | 183050.79 | 44753.76 | −550942.26 | −788639.00 | −519677.33 |
| 6 | 674064.17 | 405102.5 | 746106.31 | 268961.67 | 0.6302 | 424774.77 | 255283.29 | 169491.48 | 31264.92 | −519677.33 | −743885.24 | −474923.57 |
| 7 | 674064.17 | 405102.5 | 1015067.98 | 268961.67 | 0.5835 | 393309.97 | 236373.42 | 156936.55 | 21841.64 | −474923.57 | −679823.06 | −410861.39 |
| 8 | 674064.17 | 405102.5 | 1284029.65 | 268961.67 | 0.5403 | 364175.90 | 218864.28 | 145311.62 | 15258.54 | −410861.39 | −588122.10 | −319160.43 |
| 9 | 674064.17 | 405102.5 | 1552991.32 | 268961.67 | 0.5002 | 337199.90 | 202652.11 | 134547.80 | 10659.60 | −319160.43 | −456857.97 | −187896.30 |
| 10 | 674064.17 | 405102.5 | 1821952.99 | 268961.67 | 0.4632 | 312222.13 | 187640.84 | 124581.29 | 7446.78 | −187896.30 | −268961.67 | 0.00 |
| Σ | | | | | | 3898891.96 | 2897529.58 | 1001362.37 | 0.00 | | | |
| NPV | 1001362.37 | | | | | | | | | | | |
| BCR | 1.346 | | | | | | | | | | | |
| IRR | 0.4314 | | | | | | | | | | | |
| RIC | 0.43143 | | | | | | | | | | | |

*Source: Calculation was based on equ-2, 3 & 4.*

for reinvestment, RIC was found close to the IRR and more than δ = 8%, also showing that the investment option in the *B. ceiba* production business was profitable.

Table 5 contained the results of NPV of *W. somnifera*. After 10th year the NPV was found positive (BDT 519378.28), BCR was observed more than unity (1.21), and IRR was shown more than zero (40.83%). These results indicated that *W. somnifera* production in the arable land was profitable even in the long-run. Taking into consideration reinvestment

**Table 5. Cash-flow streams and estimation of NPV, BCR, IRR and RIC for *W. somnifera* production (BDT./ha).**

| t | Bt | Ct | Cash balance | Nt | Discount Factor | Discounted Bt | Discounted Ct | Discounted Nt | Nt at IRR | PBt-1 | (1+?)*PBt-1 | PBt |
|---|---|---|---|---|---|---|---|---|---|---|---|---|
| 1 | 0 | 335236.25 | −335236.25 | −335236.25 | 0.9259 | 0.00 | 310403.94 | −310403.94 | −238044.84 | 0.00 | 0.00 | −335236.25 |
| 2 | 524408.09 | 380950.29 | −191778.45 | 143457.80 | 0.8573 | 449595.41 | 326603.47 | 122991.94 | 72333.54 | −335236.25 | −472109.98 | −328652.18 |
| 3 | 524408.09 | 380950.29 | −48320.65 | 143457.80 | 0.7938 | 416292.05 | 302410.62 | 113881.43 | 51362.66 | −328652.18 | −462837.70 | −319379.90 |
| 4 | 524408.09 | 380950.29 | 95137.15 | 143457.80 | 0.7350 | 385455.60 | 280009.84 | 105445.77 | 36471.64 | −319379.90 | −449779.64 | −306321.84 |
| 5 | 524408.09 | 380950.29 | 238594.95 | 143457.80 | 0.6806 | 356903.33 | 259268.37 | 97634.97 | 25897.81 | −306321.84 | −431390.10 | −287932.30 |
| 6 | 524408.09 | 380950.29 | 382052.75 | 143457.80 | 0.6302 | 330466.05 | 240063.30 | 90402.75 | 18389.54 | −287932.30 | −405492.29 | −262034.49 |
| 7 | 524408.09 | 380950.29 | 525510.55 | 143457.80 | 0.5835 | 305987.08 | 222280.84 | 83706.25 | 13058.06 | −262034.49 | −369020.64 | −225562.84 |
| 8 | 524408.09 | 380950.29 | 668968.35 | 143457.80 | 0.5403 | 283321.37 | 205815.59 | 77505.79 | 9272.28 | −225562.84 | −317657.98 | −174200.18 |
| 9 | 524408.09 | 380950.29 | 812426.15 | 143457.80 | 0.5002 | 262334.61 | 190569.99 | 71764.62 | 6584.07 | −174200.18 | −245324.43 | −101866.63 |
| 10 | 524408.09 | 380950.29 | 955883.95 | 143457.80 | 0.4632 | 242902.41 | 176453.69 | 66448.72 | 4675.22 | −101866.63 | −143457.80 | 0.00 |
| Σ | | | | | | 3033257.92 | 2513879.64 | 519378.28 | 0.00 | | | |
| NPV | 519378.28 | | | | | | | | | | | |
| BCR | 1.207 | | | | | | | | | | | |
| IRR | 0.4083 | | | | | | | | | | | |
| RIC | 0.40829 | | | | | | | | | | | |

*Source: Calculation was based on equ-2, 3 & 4.*

possibilities, RIC was found in close to proximity to the IRR and more than δ = 8%, which also clearly indicated the reinvestment option of *W. Somnifera*.

## The Net-Present-Value under risks of medicinal plants production

Medicinal plant producers face considerable financial uncertainty due to unpredictable production levels and market demand. To account for these fluctuations, key variables—namely yield, producer price, seed cost, and market absorption—were modeled as stochastic inputs within a Monte Carlo simulation framework. Table 6 summarizes the minimum, median, and maximum values of these parameters for *A. vera*, *B. ceiba*, and *W. somnifera*, providing a quantitative view of the risk landscape for producers. These observed ranges are consistent with earlier research highlighting the variability of non-conventional crops (4, 14, 21, 36] in the South Asian context [29,38,78,79].

According to Table 6, *A. vera* demonstrates the greatest yield potential (23,197–56,884 kg/ha) but is particularly vulnerable to price swings (Tk. 19–50/kg). *B. ceiba* generates moderate yields (5,196–12,152 kg/ha), with profitability largely influenced by seed price variability (Tk. 117.6–250/unit). In contrast, *W. somnifera* shows lower yield levels (1,081–3,742 kg/ha) but achieves premium market prices (Tk. 120–308/kg), although the high cost of planting material (Tk. 1,000–2,200/unit) constrains expansion. Across all three species, market absorption ranged between 65% and 100%, indicating potential oversupply risks during peak harvests. Collectively, these findings emphasize that variability in yield, price, planting material, and market uptake constitutes a major limitation to the long-term viability of medicinal plant cultivation.

By incorporating these stochastic input parameters into the simulation model, NPVs for medicinal plant production were estimated. The resulting distributions are depicted in Fig 4, while Table 6 presents the descriptive statistics of each indicator generated from the Monte Carlo simulations.

Frequency distribution of simulated NPVs ($\widetilde{NPV}$) for *A. vera*, *B. ceiba* and *W. somnifera* production was shown in Fig 3. Fig 3 showed that the data followed a normal distribution. The study also tested the normality of data distribution. Table 7 showed that the Jarque-Bera (0.0799, 0.0621, 0.0518) and Shapiro-Wilk (0.1272, 0.1104, 0.9823) probabilities were both above 0.05 (traditional alpha level), indicating that the data followed a normal distribution.

Table 7 also showed that the impact of changes in price, yield and market absorption was more severe on the NPV of *A. vera, B. ceiba* and *W. somnifera* farms. In this case, the minimum NPV range was found at negative values for a hectare of plants, which means that MP producers could have suffer long-term losses. But there were also opportunities to create positive value, meaning the greatest measure of benefit. The average profit value of BDT. 862686.83, BDT. 934800.82 and BDT. 455538.84, the positive value were found by *A. vera, B. ceiba* and *W. somnifera* producers during

**Table 6. Stochastic values of yield, producer price, seed cost, and market absorption for selected MPs.**

| Stochastic variables | Unit | GRKS distribution | | | | | | | | |
|---|---|---|---|---|---|---|---|---|---|---|
| | | *A. vera* | | | *B. ceiba* | | | *W. somnifera* | | |
| | | **Min** | **Max** | **Mid**♯ | **Min** | **Max** | **Mid** | **Min** | **Max** | **Mid** |
| Yield | kg/ha | 23197 | 56884 | 38594 | 5196 | 12152 | 6984 | 1081 | 3742 | 2457 |
| Producer's price | Tk*/kg | 19 | 50 | 36.71 | 34.66 | 90.33 | 57.05 | 120 | 308 | 207.4 |
| Seed price | Tk/unit | 1.5 | 9 | 5.86 | 117.6 | 250 | 192.1 | 1000 | 2200 | 1848 |
| Market absorption | % | 72 | 100 | 83 | 70 | 100 | 91 | 65 | 100 | 86 |

*1 US$ is equivalent to BDT. 86.10 (during data analysis time).

♯GRKS parameters are Min (minimum), Mid (mid-point or middle or mode), and Max (maximum).

*Source: The stochastic risk variables were derived from surveys and field-level records.*

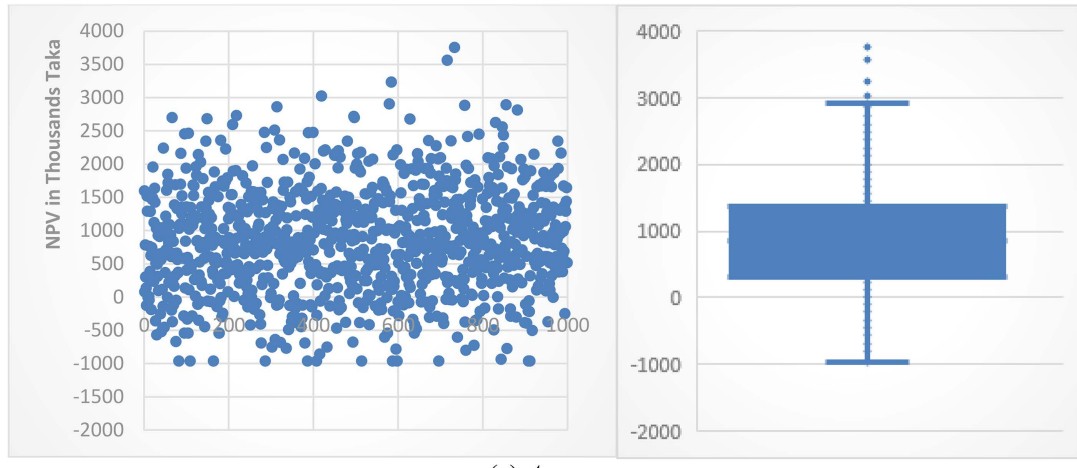

(a) *A. vera*

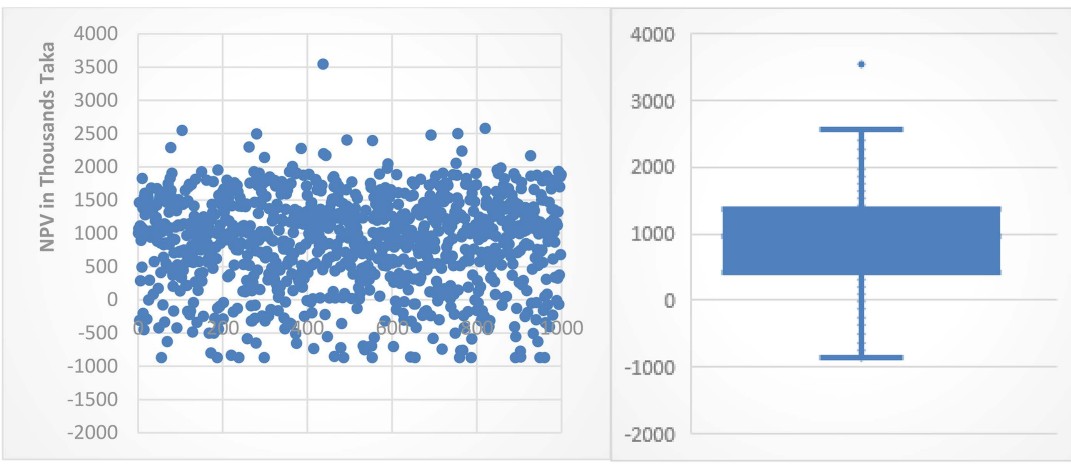

(b) *B. ceiba*

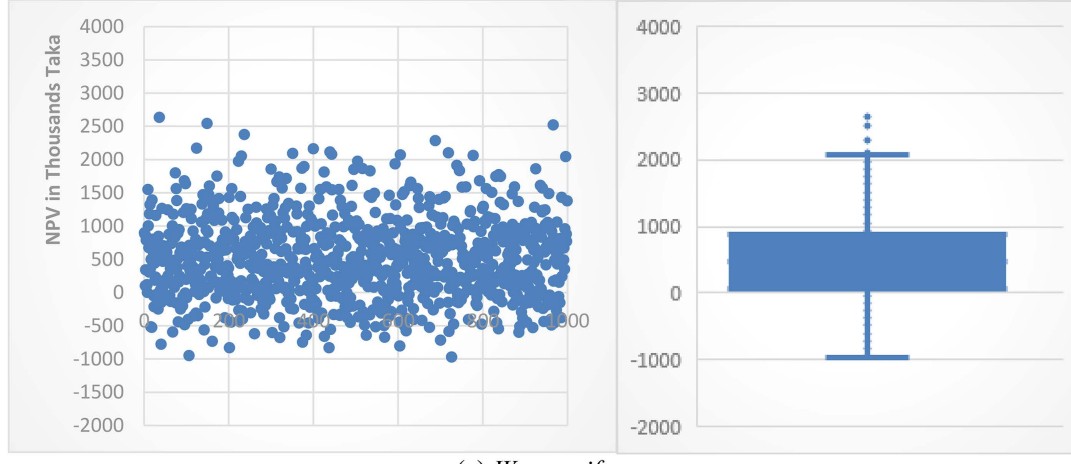

(c) *W. somnifera*

**Fig 3. Frequency distribution of NPVs for MPs (a+b+c) production.**

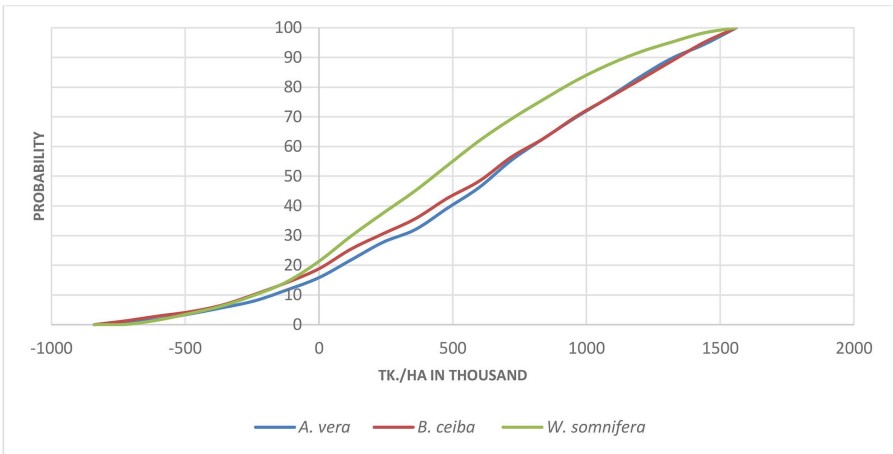

**Fig 4. CDFs of simulated NPVs.**

**Table 7. Summary statistics of Monte-Carlo simulation for Net-Present-Value.**

| Statistic | | Aloevera | Shimulmul | Ashwagandha |
|---|---|---|---|---|
| Mean | | 862686.83 | 934800.82 | 455538.84 |
| Std. deviation | | 749862.60 | 844512.78 | 426270.51 |
| Maximum | | 3728774.85 | 3870783.84 | 2630712.82 |
| Minimum | | −1288129.93 | −1518492.53 | −973013.40 |
| Coefficient of Variation, CV | | 0.8692 | 0.9034 | 0.9358 |
| Probability of NPV < 0 | | 0.1320 | 0.1540 | 0.1990 |
| Confidence interval for 0.05 level of significance in one-tail test | : Lower value | 917937.50 | 997025.43 | 486946.91 |
| | : Upper value | 807436.15 | 872576.22 | 424130.77 |
| Jarque-Bera | | 0.0799 | 0.0621 | 0.0518 |
| **Shapiro-Wilk** | | 0.1272 | 0.1104 | 0.9823 |
| p-value of **t-test (with** $H_0$: $\mu \le 0$) | | 0.000*** | 0.000*** | 0.000*** |

*Source: Calculation was based on equ-5, 6 & 7.*

the agricultural cycle, respectively. The coefficient of variation (CV) indicating the relative variability at 86.92%, 90.34% and 93.58% for *A. vera*, *B. ceiba* and *W. somnifera*, respectively.

Fig 4 showed the cumulative distribution functions for *A. vera, B. ceiba* and *W. somnifera* production practices, where x-axis represents the simulated NPVs in thousands of taka, and the y-axis represents the probability of that NPV. Fig 5 also showed that the CDF increases monotonically and continuously on the left and sign boundaries. This indicated that the producer will benefit from *A. vera, B. ceiba* and *W. somnifera* production in the long run under the risks of price, yield and market absorption. Exceeding the break-even point can be expected with a risk of 13.2%, 15.4% and 19.9%, for selected MP, respectively.

## The Real-Option-Value of the medicinal plants production

In this study, ROV was evaluated by constructing a binomial decision tree. Given the maturity time (years: T = 10) and the number of steps (n = 10), the nodes of the decision tree were filled with the values. Here, the output of *A. vera*, PV at t = 0

was estimated at BDT. 1034136. Thus, NPV = PV + IE = 1034136 + 0 = 1034136. For year-1, the upper and lower values were calculated using annual volatility (0.1285) of *A. vera* production, therefore, the NPV for year-1 was found to be:

- Upper: $(PV_{t-1} * e^{(\sigma_z \sqrt{dt})})$ = BDT 1175938
- Lower: $(PV_{t-1} * e^{(-\sigma_z \sqrt{dt})})$ = BDT 909433

Where, $dt = 1$ (Fig 5).

The study then calculated NPV for year-2 using the same annual volatility, which yielded the upper value at BDT. 1337185 and lower value at BDT. 1034136, based on the upper value of BDT. 1175938 (of year-1). It was also found that

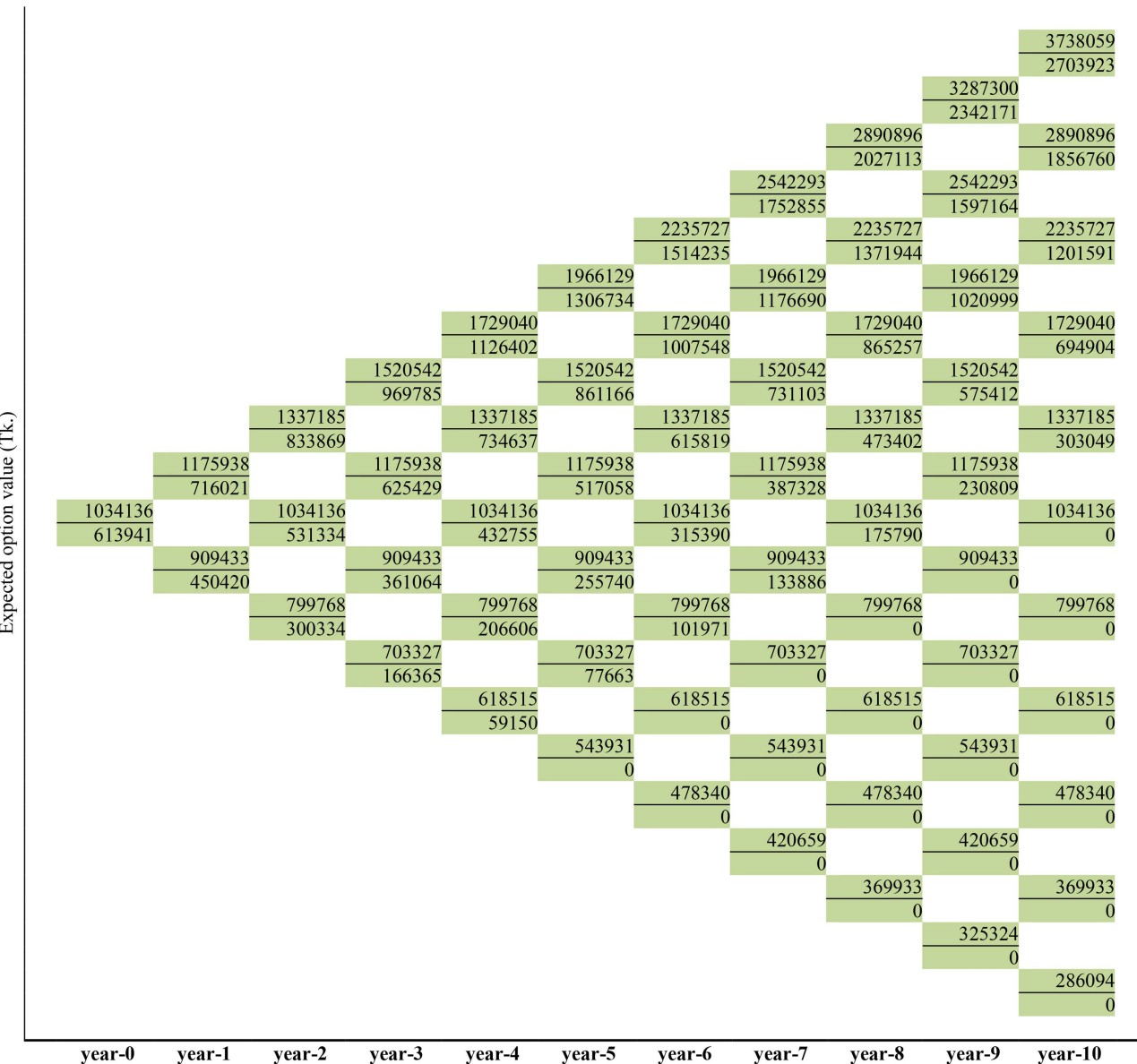

**Fig 5. Lattices depicting the first-year dynamics of project value, the value of option and the decision for *A. vera* production.**

the lower value of BDT. 1175938 (year-1) and the upper value of BDT. 909433 (year-1) was same, that is BDT. 1034136, ensures a recombining lattice. This procedure was followed for all the years until calculated NPVs for all nodes in the decision tree (upper values of all nodes in Fig 5). After that, the study determined the option value for *A. vera* production.

As described in Copeland and Antikarov [86] cited by Asci et al. [82], this study analyzes the optimal execution of real options starting from the end of the tree when the option expires. The final node of the options calculation is chosen as the maximum of the two values, expressed as $MAX\ [0,\ \{NPV^{av}(at\ t = 10) - (ePV^{av} - IE^{av})\ (at\ t = 0)\}]$. Therefore, the top node at the end of the three is filled with the maximum value of BDT. 2703923 comparing with the binomial tree as shown in Fig 6. The remaining nodes at t = 10 were calculated by repeating the portfolio approach. The study then discounted the

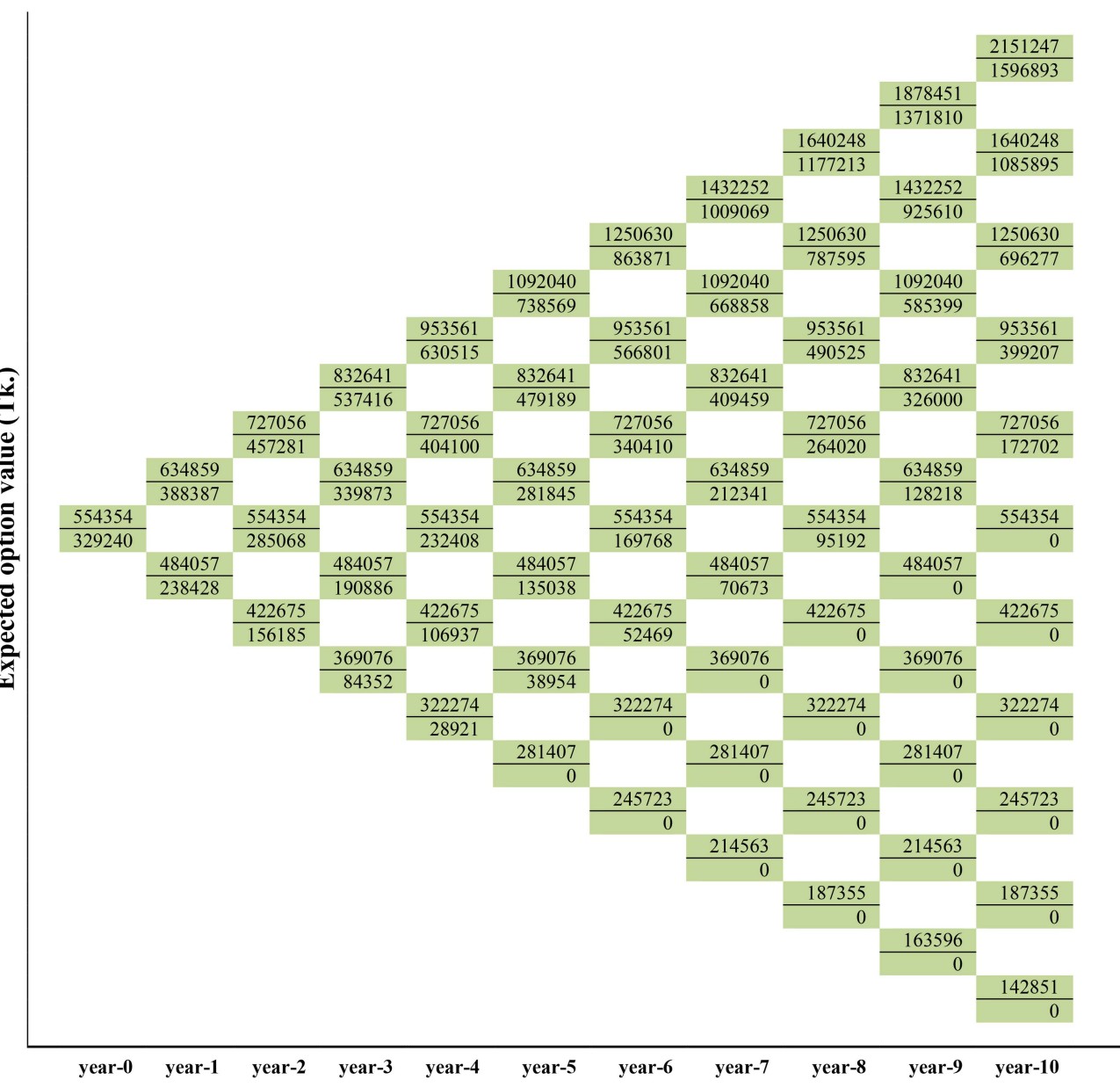

**Fig 6. Lattices depicting the first-year dynamics of project value, the value of option and the decision for *B. ceiba* production.**

payoffs back to date. This means stepping back through the lattice and calculating the option price at each point. This is done by the equation $V_n = e^{-r\Delta t}(p\,V_u + (1-p)\,V_d)$, where $\Delta t$ = time step, p = probability of stock increasing by a factor u, and 1-p = probability of stock decreasing by a factor d. Hence, the value of the option is found to be BDT. 2342171 where t = 9. The study repeated the process for all remaining nodes. By replicating the portfolio approach above, the NPV of the *B. ceiba* and *W. somnifera* options was calculated accordingly (Figs 5 and 7).

Finally, the expected NPV with option value (eNPV) was calculated by simply adding the option value to the static NPV. The NPV with static and optional values is shown in Table 8.

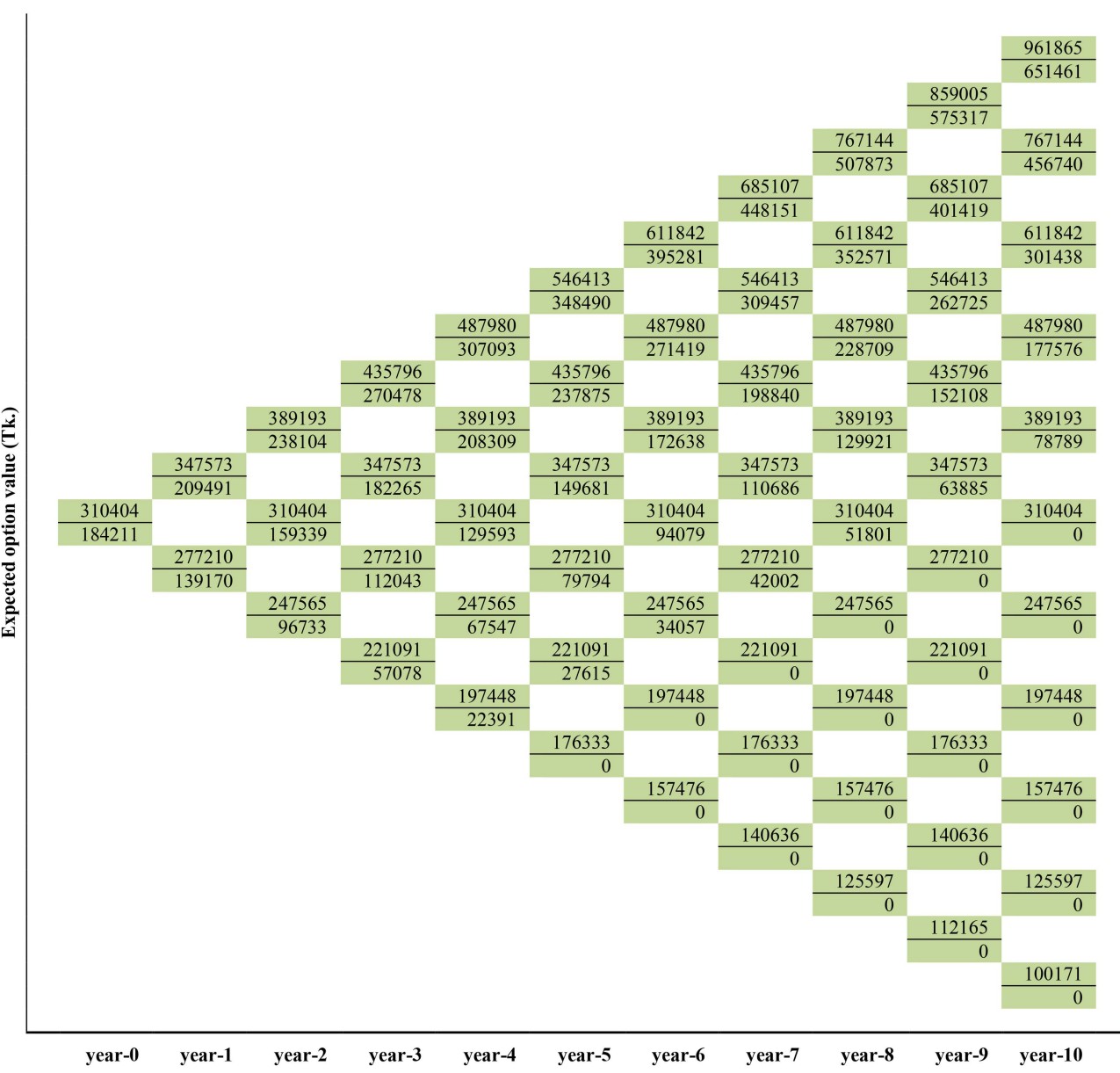

**Fig 7. Lattices depicting the first-year dynamics of project value, the value of option and the decision for *W. somnifera* production.**

**Table 8. NPV with option value for _A. vera_, _B. ceiba_ and _W. somnifera_ production (BDT./ha).**

| Medicinal Plant | NPV | Binomial Option Value | | eNPV | Percentage of Change |
|---|---|---|---|---|---|
| Aloevera | 1009546.16 | Call | 613940.88 | 1623487.04 | 60.81% |
| | | Put | 253.05 | 1009799.21 | 0.03% |
| Shimulmul | 1001362.37 | Call | 329239.99 | 1330602.36 | 32.88% |
| | | Put | 269.66 | 1001632.03 | 0.03% |
| Ashwagandha | 519378.28 | Call | 184210.91 | 703589.20 | 35.47% |
| | | Put | 7.80 | 519386.08 | 0.00% |

_Source: Calculation was based on equ-8, 9, 10, 11 & 12._

The results in Table 8 showed that the NPV with the option value (call value) was BDT. 1009546 + 613941 = 1623487 for _A. vera_ production. It was also estimated at BDT 1330602 and BDT. 703589 for the production of _B. ceiba_ and _W. somnifera_, respectively. It was assumed that, once a farmer starts producing MP, it would take a time of 10 years to establish a market position and gain a reputation as a producer of MP. This market position was close to the present value of future MP premiums, which was estimated at the option value. The results in Table 8 showed that the option value of the decision to expand production of _A. vera_ increased the NPV by 61%, compared to 33% and 35% for _B. ceiba_ and _W. somnifera_ production, respectively. This result clearly indicated that the investment in _A. vera, B. ceiba_ and _W. somnifera_ production was viable in the long run. Table 8 also resulted in the finding that _A. vera_ is more likely to pay future premiums than the production of _B. ceiba_ and _W. somnifera_.

## Discussion

Medicinal plants are among Bangladesh's most valuable natural resources. Because of its tropical climate and rich soil, Bangladesh is home to a sizable number of medicinal plants. Approximately 5700 plant species are thought to exist, with 1,000 of them thought to have therapeutic qualities [88]. There are 747 medicinal plants in Bangladesh [89]. The majority of medicinal plants are widely utilized to make _Homeopathic, Ayurvedic,_ and _Unani_ medications. Additionally, these plants are essential raw ingredients for a variety of contemporary treatments. In the last decade, the herbal industry has been dependent on the wild collection of MP from the forest. As a result, due to irresponsible exploitation and the nation's expanding population, these valuable resources have been gradually depleted. The demand for MP gradually increased over the year, and hence the price of MP increased which gave scope for commercial cultivation. But the business sustainability of MP is not clear. Due to the potential domestic and international market for MPs, commercial production of these plants is expanding in Bangladesh. It is suggested that commercial farming could offer local residents a viable alternative and long-term means of subsistence while also preserving MPs' natural resource base, given the market potential and the deficiency from natural sources [67]. However, very little is known about MPs, particularly about their business viability, policy needs, and research questions in relation to Bangladesh.

In Bangladesh, MP production in crops field is profitable which is showed by the findings of this study. These results aligned with a study conducted by Chowdhury et al. [8] where the authors found that _A. vera_ grown in calcareous soils yielded a fresh leaf gel weight of 907 g per plant, suggesting that soil type significantly affects yield and, consequently, profitability [88]. Both the current study and previous research indicated that _A. vera_ cultivation is financially viable in Bangladesh. The positive NPV, BCR, and IRR in this study corroborate earlier findings of profitability. Additionally, the impact of soil types on yield underscores the importance of site selection for optimizing returns. Overall, the results showed that MP production businesses were sustainable in the long term. Of the three MPs, _B. ceiba_ showed better option in the long run.

The results of simulated NPV ($\widetilde{NPV}$) for *A. vera, B. ceiba* and *W. somnifera* production under embedded yield, price and market risk showed that, although the positive values, there was also a 13.2% possibility to loss. Similarly, the probability of negative NPV was 15.4% for *B. ceiba* and 19.9% for *W. somnifera*. The higher level of risk suggests that MP production businesses might not have been the best long-term choice, and the farmers could have been disappointed with MP production because the investment costs exceed the return on investment. In this case, ROV could have provided accurate decisions about MP production alternatives.

The current study's analysis of risk factors corroborates the high risk-bearing capacity highlighted in previous research [90]. Goswami et al. [91] developed an efficient in vitro regeneration system for *W. somnifera*, which could enhance propagation and potentially increase yields [90]. A study on agroforestry systems in Charland, Bangladesh, found that *W. somnifera* in sole cropping had the highest dry root yield of 692.0933 kg/ha, indicating that cultivation practices significantly affect profitability [92]. The government does not prioritize or even acknowledge the potential of therapeutic plants as a source of income. However, the current study's results indicate that the start of commercial MP production may be a positive step in the effort to steer local people toward alternative sources of income. Business sustainability of commercial cultivation of MP under risks also sustainable in the long run.

This study is based on primary data collection from the two commercially cultivated MP districts (Natore and Bogura) in Bangladesh. A small sample is used in this study to examine the real situation in and around the commercially cultivated areas. In Bangladesh's other commercially cultivated regions, different outcomes can be observed. It is strongly advised that more research be done to reexamine the current findings. The future research could deal with more areas and also compare the results with wild collections of MP and investigate business sustainability considering other risks.

## Conclusion and policy recommendations

This study assessed the business sustainability of medicinal plant (MP) production-specifically *A. vera, B. ceiba* and *W. somnifera* – under risk conditions by analyzing three key economic indicators: static NPV (including BCR, IRR, and RIC), risk-adjusted simulated NPV ($\widetilde{NPV}$), and real option value (ROV). The analytical results of first economic indicators showed that the MP production is financially sound and sustainable in the long-run. On the other hand, after incorporating risk into the simulation model ($\widetilde{NPV}$) the results indicate that MP producers are likely suffering losses even in the long-run. But, considering the time dynamics for MP production, the NPV with the value of the option is found at increased value. Based on these research findings, this study concluded that investment in MP production business at the farm level is profitable even if there are some risks and uncertainties. Therefore, further expansion of the MP production area can contribute to the business sustainability in rural areas and foster their development. But caution should be taken about the food security issue, as more allocation of areas may hamper the production of cereal crops (rice) which is the main food of Bangladesh.

## Supporting information

**S1 File. Data and analysis.**
(XLSX)

**S2 File. Questionnaire.**
(DOCX)

**S3 File. Supporting information.**
(DOCX)

## Acknowledgments

We would also like to express our appreciation to the farmers, who provide information for writing up the manuscript.

# Author contributions

**Conceptualization:** Md. Abu Saiyem, Mst. Esmat Ara Begum, Mohammad Ismail Hossain.

**Data curation:** Md. Abu Saiyem, Mst. Fatema Begum.

**Formal analysis:** Md. Abu Saiyem.

**Investigation:** Mst. Fatema Begum, Mst. Esmat Ara Begum.

**Methodology:** Md. Abu Saiyem, Mst. Esmat Ara Begum, Mohammad Ismail Hossain.

**Software:** Md. Abu Saiyem.

**Supervision:** Mohammad Ismail Hossain.

**Validation:** Md. Abu Saiyem.

**Visualization:** Mst. Fatema Begum.

**Writing – original draft:** Md. Abu Saiyem.

**Writing – review & editing:** Mst. Esmat Ara Begum, Mohammad Ismail Hossain.

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
