## [Decision Letter · Decision Letter 0]

14 May 2025

PONE-D-25-10568Business Sustainability of Medicinal Plants Production under Risk in the Northwest Region of Bangladesh: A Simulation AnalysisPLOS ONE

Dear Dr. Hossain,

Thank you for submitting your manuscript to PLOS ONE. After careful consideration, we feel that it has merit but does not fully meet PLOS ONE’s publication criteria as it currently stands. Therefore, we invite you to submit a revised version of the manuscript that addresses the points raised during the review process.

We look forward to receiving your revised manuscript.

Kind regards,

Awatif Abid Al-Judaibi, PhD

Academic Editor

PLOS ONE

2. You indicated that ethical approval was not necessary for your study. We understand that the framework for ethical oversight requirements for studies of this type may differ depending on the setting and we would appreciate some further clarification regarding your research. Could you please provide further details on why your study is exempt from the need for approval and confirmation from your institutional review board or research ethics committee (e.g., in the form of a letter or email correspondence) that ethics review was not necessary for this study? Please include a copy of the correspondence as an ""Other"" file.

Reviewers' comments:

Reviewer's Responses to Questions

**Comments to the Author**

1. Is the manuscript technically sound, and do the data support the conclusions?

Reviewer #1: Partly

Reviewer #2: Yes

2. Has the statistical analysis been performed appropriately and rigorously? 

Reviewer #1: Yes

Reviewer #2: Yes

3. Have the authors made all data underlying the findings in their manuscript fully available?

Reviewer #1: Yes

Reviewer #2: Yes

4. Is the manuscript presented in an intelligible fashion and written in standard English?

Reviewer #1: Yes

Reviewer #2: Yes

5. Review Comments to the Author

Reviewer #1: The manuscript id: PONE-D-25-10568 entitled:

"Business Sustainability of Medicinal Plants Production under Risk in the Northwest Region of Bangladesh: A Simulation Analysis" has been reviewed. In my opinion there is no problem with its scientific content but I think this manuscript subject is not qualified to be published in the PLOS ONE but you are encouraged to submit this paper to another Journal which better covers the subject.

Best regards

Reviewer #2: This is an interesting study, which reports the simulation analysis regarding the sustainability of medicinal plant business in the northwest Bangladesh. Overall, the study is well-organized and well-written. However, following are some aspects, which need to be addressed for improved presentation, scientific rigor and validity of the manuscript.

Abstract: Please revise the last sentence.

Introduction: It’s a bit lengthy. Shorten the introduction.

Introduction should not be starting locally from Bangladesh. Remove the first paragraph and use some of its part to be presented as just 1-2 sentences at the end of second paragraph.

Its better to not start sentence with an abbreviation (throughout the manuscript).

L 97. Please remove grammatical errors from the manuscript.

L 100-107. Revise to present a single sentence by stating the objective of this study instead of numbering the research questions.

Remove L 108-110.

Methodology: L 114. Sustainability “of” instead of “in”

Fig 2. As the study areas are restricted to two yellow points, is this a true representation of the complete northwest region?

Please use past tense while describing the methodology of the currents study.

L 168. Please revise this sub-heading

L 205. Why no approval of any regulatory/local law authority or research organization/institution was obtained to conduct this study?

Although no human subjects were directly involved, the study focused on medicinal plants business and publication of these findings will be directly associated with the business related to human and animal health. Therefore, at least permission to conduct this study should be mentioned with relevant approval number from the relevant institution/ regulatory authority for the validity of the study.

Table 1. How it was decided to use these assumptions, As these values may affect the outcomes, it is important to mention that how these assumption values including interest and discount rates were decided.

L 267. On which basis authors calculated these values. Please indicate the relevant sources of values used for calculation.

The results of present study should be presented in past tense. Please check complete manuscript for grammatical mistakes according to each section.

Another but major concern regarding results and discussion section is the lack of comparison of the current results with previous studies conducted in Bangladesh, neighboring countries, or anywhere around the world, which should be incorporated with relevant explanations of any differences in findings.

Tables. Please add the basis of the calculations with relevant references in methodology and remove the footnote of authors own calculation from result tables.

Revise the first sentence of conclusion.

Avoid using abbreviations in headings, table and figure captions.

6. PLOS authors have the option to publish the peer review history of their article (what does this mean? ). If published, this will include your full peer review and any attached files.

**Do you want your identity to be public for this peer review?** For information about this choice, including consent withdrawal, please see our Privacy Policy .

Reviewer #1: No

Reviewer #2: **Yes: ** Muhammad Uzair Akhtar

---

## [Author Response · Author response to Decision Letter 1]

18 Jun 2025

We upload a file named as response to reviewers. We confirm that we made revision based on the comments and suggestions raised by the reviewers.

---

## [Decision Letter · Decision Letter 1]

14 Jul 2025

PONE-D-25-10568R1Business sustainability of medicinal plant production under risk in the northwest region of Bangladesh: a simulation analysisPLOS ONE

Dear Dr. Hossain,

Thank you for submitting your manuscript to PLOS ONE. After careful consideration, we feel that it has merit but does not fully meet PLOS ONE’s publication criteria as it currently stands. Therefore, we invite you to submit a revised version of the manuscript that addresses the points raised during the review process.

We look forward to receiving your revised manuscript.

Kind regards,

Awatif Abid Al-Judaibi, PhD

Academic Editor

PLOS ONE

Journal Requirements:

Reviewers' comments:

Reviewer's Responses to Questions

**Comments to the Author**

1. If the authors have adequately addressed your comments raised in a previous round of review and you feel that this manuscript is now acceptable for publication, you may indicate that here to bypass the “Comments to the Author” section, enter your conflict of interest statement in the “Confidential to Editor” section, and submit your "Accept" recommendation.

Reviewer #3: (No Response)

Reviewer #4: (No Response)

2. Is the manuscript technically sound, and do the data support the conclusions?

Reviewer #3: Partly

Reviewer #4: Yes

3. Has the statistical analysis been performed appropriately and rigorously? 

Reviewer #3: Yes

Reviewer #4: Yes

4. Have the authors made all data underlying the findings in their manuscript fully available?

Reviewer #3: No

Reviewer #4: Yes

5. Is the manuscript presented in an intelligible fashion and written in standard English?

Reviewer #3: Yes

Reviewer #4: Yes

6. Review Comments to the Author

Reviewer #3: This is an interesting manuscript, a good research question, sound methodology, and comprehensive research design. The authors have improved the manuscript based of first reviewers' reports (I was not a part of the first review round).

My main concern with the manuscript is that it is very unclear how the 196 interviews, which were mentioned as the primary source of data in the abstract, relate to the presented findings. There is limited or no information on the interview design (which questions were asked) nor on the interview results (which answers were given). Most of the modeling input data is attributed to secondary sources rather than the interviews themselves. I would expect a wide range of interview responses based on these 196 interviews with individual farms. This would require a dedicated Results section where these descriptive results are presented, and consequently, how they are then used (aggregated?) in the general modeling exercise. If modeling was mainly done based on other sources, the manuscript methodology and abstract should be written to reflect so. If 196 interviews were performed using the same interview design and protocol, these results would be of much interest to the academic community and should thus be given more attention. For example, were your three main risk indicators qualified based on the farmer interviews or were they selected a priori? Etc.

A minor suggestion would be to move all the discussion of results that are currently integrated in the Results section to a dedicated Discussion section, rather than a Conclusion.

Reviewer #4: This paper presents a technically rigorous study that explores an important issue in agricultural sustainability. The authors used a scientific methodology, combining Monte Carlo simulations and ROV, to assess the commercial sustainability of medicinal plant production under uncertainty. The data source is reliable, coming from a valid survey of 197 farmers, and supports the conclusions presented in the paper. The statistical analysis is rigorous, and the advanced methods used add credibility to the findings.

Although the paper is well presented and written in standard English, some minor areas could be improved. The introduction section could be reorganized and optimized to present the research question more clearly. In addition, the literature review should be further expanded and presented as a separate section after the introduction section, with an additional discussion of the research gaps in existing research and an emphasis on the theoretical contribution of the study.

Finally, the discussion section is weak and could be improved by a deeper extension of the analysis of the findings, their significance for agricultural development in countries including Bangladesh, possible policy implications, and future research directions. Strengthening these sections would enhance the theoretical and practical contributions of the paper.

Overall, the paper is technically rigorous, the statistical analysis is appropriate, and the data support the conclusions. The paper is clear, but further development in some areas would significantly improve its clarity and impact.

7. PLOS authors have the option to publish the peer review history of their article (what does this mean? ). If published, this will include your full peer review and any attached files.

**Do you want your identity to be public for this peer review?** For information about this choice, including consent withdrawal, please see our Privacy Policy .

Reviewer #3: No

Reviewer #4: No

---

## [Author Response · Author response to Decision Letter 2]

19 Jul 2025

Answer: Thank you for your suggestions.

2. Is the manuscript technically sound, and do the data support the conclusions?

Reviewer #3: Partly

Reviewer #4: Yes

Answer: Thanks the both reviewers for their thoughtful comments and suggestions.

3. Has the statistical analysis been performed appropriately and rigorously?

Reviewer #3: Yes

Reviewer #4: Yes

Answer: Thanks

4. Have the authors made all data underlying the findings in their manuscript fully available?

Reviewer #3: No

Reviewer #4: Yes

Answer: We uploaded the data file in the submission process.

5. Is the manuscript presented in an intelligible fashion and written in standard English?

Reviewer #3: Yes

Reviewer #4: Yes

Answer: Thanks

Reviewer #3: This is an interesting manuscript, a good research question, sound methodology, and comprehensive research design. The authors have improved the manuscript based of first reviewers' reports (I was not a part of the first review round).

Answer: Thank you for your appreciation.

My main concern with the manuscript is that it is very unclear how the 196 interviews, which were mentioned as the primary source of data in the abstract, relate to the presented findings. There is limited or no information on the interview design (which questions were asked) nor on the interview results (which answers were given). Most of the modeling input data is attributed to secondary sources rather than the interviews themselves. I would expect a wide range of interview responses based on these 196 interviews with individual farms. This would require a dedicated Results section where these descriptive results are presented, and consequently, how they are then used (aggregated?) in the general modeling exercise. If modeling was mainly done based on other sources, the manuscript methodology and abstract should be written to reflect so. If 196 interviews were performed using the same interview design and protocol, these results would be of much interest to the academic community and should thus be given more attention. For example, were your three main risk indicators qualified based on the farmer interviews or were they selected a priori? Etc.

Answer: Thanks for your concern about the interview and the questions. This manuscript is based on the survey questionnaire. We used pre-tested questionnaire for collecting necessary data from the respondents. The questionnaire is attached in the supporting file. You may look what types of questions are asked during the survey. The questionnaire is transferred into the excel sheet. The data file is also found in the submission system. In the modelling purpose we take few assumptions that also explained in the previous version. For your more understanding, we also present them here:

These assumptions allowed for a standardized and realistic evaluation of long-term returns and risks associated with MP production using NPV methodology:

Farm size: The NPV analysis was based on a standard 1-hectare farm, ensuring consistency across different farm types and simplifying comparison of economic outcomes.

Ownership: It was assumed that the farmer owned 100% of the land, implying no rental or lease costs were considered in the model.

Loan length: The analysis assumed a one-year loan term, reflecting short-term borrowing often used for operational capital in agriculture.

Flat rate of interest: A fixed annual interest rate of 9% was applied to borrowed capital, representing prevailing rural credit rates during the study period. This is also Bank interest rate prevailing during the data collection period.

Initial equity: It was assumed that farmers had no initial equity, meaning the entire investment was financed through credit, to simulate a conservative scenario.

Life cycle: A 10-year investment horizon was considered for the analysis, matching the productive life cycle of perennial MP species.

Discount rate: An 8% discount rate was used to reflect the time value of money and the opportunity cost of capital in agricultural investments. The discounting factor is related to the interest rate and hence it is considered 8%.

Sources of Stochastic Risk Variables Used in the Simulation Model:

Yield: Field survey data (2018-2020); validated against regional agricultural statistics and previous MP yield studies (e.g., [28], [60], [61]).

Producer's price: Local market surveys, trader interviews, and farm gate price reports collected during field visits (2018-2020); supported by Bangladesh Bureau of Statistics (BBS) and Ministry of Agriculture (MoA) reports [60], [61], [70].

Seed price: Data from local farmers, input suppliers, and Agricultural Extension Offices (UAO) reports; triangulated with previous studies on input costs [28], [34], [60].

Market absorption: Estimated from farmer responses on post-harvest sales volumes and price realization; confirmed via interviews with local buyers and cooperatives [28], [34], [60].

The three indicators used here is based on priori experience and reality that farmer faced in their farming.

A minor suggestion would be to move all the discussion of results that are currently integrated in the Results section to a dedicated Discussion section, rather than a Conclusion.

Answer: Thanks for the suggestions. We moved all discussion in the discussion section.

Reviewer #4: This paper presents a technically rigorous study that explores an important issue in agricultural sustainability. The authors used a scientific methodology, combining Monte Carlo simulations and ROV, to assess the commercial sustainability of medicinal plant production under uncertainty. The data source is reliable, coming from a valid survey of 197 farmers, and supports the conclusions presented in the paper. The statistical analysis is rigorous, and the advanced methods used add credibility to the findings.

Answer: Thank you for your positive feedback.

Although the paper is well presented and written in standard English, some minor areas could be improved. The introduction section could be reorganized and optimized to present the research question more clearly. In addition, the literature review should be further expanded and presented as a separate section after the introduction section, with an additional discussion of the research gaps in existing research and an emphasis on the theoretical contribution of the study.

Answer: The introduction is slightly modified following your comments and suggestions.

We incorporated literature review in the introduction section which is well practiced in scientific arena. At this stage, we think the introduction section is well organized and clearly present the research gaps. Therefore, we keep them as together.

Finally, the discussion section is weak and could be improved by a deeper extension of the analysis of the findings, their significance for agricultural development in countries including Bangladesh, possible policy implications, and future research directions. Strengthening these sections would enhance the theoretical and practical contributions of the paper.

Answer: We improved the discussion section with following your suggestions.

Overall, the paper is technically rigorous, the statistical analysis is appropriate, and the data support the conclusions. The paper is clear, but further development in some areas would significantly improve its clarity and impact.

Answer: Thanks for your feedback and time spent for reading this manuscript.

---

## [Decision Letter · Decision Letter 2]

30 Jul 2025

PONE-D-25-10568R2Business sustainability of medicinal plant production under risk in the northwest region of Bangladesh: a simulation analysisPLOS ONE

Dear Dr. Mohammad Ismail Hossain,

Thank you for submitting your manuscript to PLOS ONE. After careful consideration, we feel that it has merit but does not fully meet PLOS ONE’s publication criteria as it currently stands. Therefore, we invite you to submit a revised version of the manuscript that addresses the points raised during the review process.

We look forward to receiving your revised manuscript.

Kind regards,

Awatif Abid Al-Judaibi, PhD

Academic Editor

PLOS ONE

Journal Requirements:

Reviewers' comments:

Reviewer's Responses to Questions

**Comments to the Author**

1. If the authors have adequately addressed your comments raised in a previous round of review and you feel that this manuscript is now acceptable for publication, you may indicate that here to bypass the “Comments to the Author” section, enter your conflict of interest statement in the “Confidential to Editor” section, and submit your "Accept" recommendation.

Reviewer #3: (No Response)

Reviewer #4: All comments have been addressed

2. Is the manuscript technically sound, and do the data support the conclusions?

Reviewer #3: Partly

Reviewer #4: Yes

3. Has the statistical analysis been performed appropriately and rigorously? 

Reviewer #3: Yes

Reviewer #4: Yes

4. Have the authors made all data underlying the findings in their manuscript fully available?

Reviewer #3: Yes

Reviewer #4: Yes

5. Is the manuscript presented in an intelligible fashion and written in standard English?

Reviewer #3: Yes

Reviewer #4: Yes

6. Review Comments to the Author

Reviewer #3: I appreciate the authors response to my first review questions. And while I appreciate the substantial supporting information, including the survey documents and collected data, the manuscript itself should be able to stand on its own, containing the most important findings and results of the study without having to go into the SI. I still find an inconsistency in the structure and presentation of the manuscript. The abstract mentions the study method was data collection through 196 surveys which were then used in NPV and further analysis. However, the results of the surveys seem to be presented numerically in the Methods (Table 2, average scores on yield, prices, etc.). While these factors are inputs to the analysis, they are not part of the Methods but they are part of the study's results. They were not known before the study started. I would suggest moving these numbers to a first section in the Results, where I would also expect (as by my previous comments) to have a more substantial overview of what was found in the surveys, rather than only summary statistics. That is, you have a 15-page questionnaire instrument, used across a wide range of farmers in different geographical areas. A descriptive section based on the most relevant background information and outcome data of interest would help readers better understand and transition into the findings of the NPV.

Reviewer #4: The author has addressed my concerns. I think this paper has been well-written and can be accepted for this journal

7. PLOS authors have the option to publish the peer review history of their article (what does this mean? ). If published, this will include your full peer review and any attached files.

**Do you want your identity to be public for this peer review?** For information about this choice, including consent withdrawal, please see our Privacy Policy .

Reviewer #3: No

Reviewer #4: No

---

## [Author Response · Author response to Decision Letter 3]

1 Sep 2025

The Reviewers and Editor comments are attached in the response to reviewer file.

---

## [Decision Letter · Decision Letter 3]

18 Sep 2025

Business sustainability of medicinal plant production under risk in the northwest region of Bangladesh: a simulation analysis

PONE-D-25-10568R3

Dear Dr. Mohammad Ismail Hossain,

We’re pleased to inform you that your manuscript has been judged scientifically suitable for publication and will be formally accepted for publication once it meets all outstanding technical requirements.

Kind regards,

Awatif Abid Al-Judaibi, PhD

Academic Editor

PLOS ONE

Additional Editor Comments (optional):

Reviewer #3:

Reviewers' comments:

Reviewer's Responses to Questions

**Comments to the Author**

1. If the authors have adequately addressed your comments raised in a previous round of review and you feel that this manuscript is now acceptable for publication, you may indicate that here to bypass the “Comments to the Author” section, enter your conflict of interest statement in the “Confidential to Editor” section, and submit your "Accept" recommendation.

Reviewer #3: All comments have been addressed

2. Is the manuscript technically sound, and do the data support the conclusions?

Reviewer #3: Yes

3. Has the statistical analysis been performed appropriately and rigorously? 

Reviewer #3: Yes

4. Have the authors made all data underlying the findings in their manuscript fully available?

Reviewer #3: Yes

5. Is the manuscript presented in an intelligible fashion and written in standard English?

Reviewer #3: Yes

6. Review Comments to the Author

Reviewer #3: Thanks for discussing more of the primary data results in the revised manuscript. This has increased readability and comprehensiveness of the article.

7. PLOS authors have the option to publish the peer review history of their article (what does this mean? ). If published, this will include your full peer review and any attached files.

**Do you want your identity to be public for this peer review?** For information about this choice, including consent withdrawal, please see our Privacy Policy .

Reviewer #3: No

---

## [Editor Report · Acceptance letter]

PONE-D-25-10568R3

PLOS ONE

Dear Dr. Hossain,

I'm pleased to inform you that your manuscript has been deemed suitable for publication in PLOS ONE. Congratulations! Your manuscript is now being handed over to our production team.

Kind regards,

on behalf of

Professor Awatif Abid Al-Judaibi

Academic Editor

PLOS ONE